

# CPJSdraw: analysis and visualization of junction sites of chloroplast genomes

Huie Li[1], Qiqiang Guo[2], Lei Xu[3], Haidong Gao[3], Lei Liu[3] and Xiangyang Zhou[3]

[1] College of Agriculture, Guizhou University, Guiyang, Guizhou, China
[2] Institute for Forest Resources & Environment of Guizhou, Guizhou University, Guiyang, Guizhou, China
[3] Nanjing Genepioneer Biotechnologies Co., Ltd, Nanjing, Jiangsu, China

## ABSTRACT

**Background**. Chloroplast genomes are usually circular molecules, and most of them are tetrad structures with two inverted repeat (IR) regions, a large single-copy region, and a small single-copy region. IR contraction and expansion are among the genetic diversities during the evolution of plant chloroplast genomes. The only previously released tool for the visualization of junction sites of the regions does not consider the diversity of the starting point of genomes, which leads to incorrect results or even no results for the examination of IR contraction and expansion.

**Results**. In this work, a new tool named CPJSdraw was developed for visualizing the junction sites of chloroplast genomes. CPJSdraw can format the starting point of the irregular linearized genome, correct the junction sites of IR and single-copy regions, display the tetrad structure, visualize the junction sites of any number ($\geq 1$) of chloroplast genomes, show the transcription direction of genes adjacent to junction sites, and indicate the IR expansion or contraction of chloroplast genomes.

**Conclusions**. CPJSdraw is a software that is universal and reliable in analysis and visualization of IR expansion or contraction of chloroplast genomes. CPJSdraw has more accurate analysis and more complete functions when compared with previously released tool. CPJSdraw as a perl package and tested data are available at http://dx.doi.org/10.5281/zenodo.7669480 for English users. In addition, an online version with a Chinese interface is available at http://cloud.genepioneer.com:9929/#/tool/alltool/detail/335.

## INTRODUCTION

Chloroplast genomes are usually circular molecules, and most of them are tetrad structures with two inverted repeat (IR) regions, namely, IRb and IRa; a large single-copy (LSC) region and a small single-copy (SSC) region (*Francois et al., 2018*). Chloroplast genomes are highly conserved, especially in the IR region. This region is of functional significance, the presence of duplicated rRNA genes in this region could be a selective advantage that allows higher production of proteins in a short time. They also play a role in replication initiation and stalling, genetic instability, gene splicing, and so on (*Zhang et al., 2019*; *Turudić et al., 2022*). This region usually ranges from 20 kb to 27 kb, which size variation among chloroplast genomes is mainly due to the expansion and contraction of the IR

Corresponding author
Lei Xu, xul@genepioneer.com

region, leading IR boundary shifts. However, the size is greatly reduced to less than 1kb in some gymnosperm cases, but is extended in the geranium chloroplast genome, where the IR region exceeds 75 kb (*Hansen et al., 2007*).

Multiple examples of IR expansion or contraction have been reported in plants, where entire genes are moved from the single-copy region to the IR region and vice versa during evolution, resulting in the diversification of the chloroplast genomes of related species, suggesting that IR boundary shifts are dynamic (*Zhu et al., 2016*). For example, analyses of single-copy and IR boundaries indicated that the IR regions of chloroplast genomes from Arecaceae (*Chen et al., 2022*), Salicaceae (*He et al., 2022*), and other families (*Wang et al., 2022*) underwent expansion or contraction.

Examination of the genes adjacent to the junction sites of the four regions of chloroplast genomes can determine whether the IR region has expanded or contracted, reflecting one of the diversities of plant chloroplast genomes, especially for related species (*Muraguri et al., 2020*; *Chen et al., 2021a*; *Chen et al., 2021b*; *Zhang et al., 2022*). A large number of plant chloroplast genomes are stored in the GenBank database, which can be freely downloaded for comparison analysis. Linearized chloroplast genome sequences from GenBank (.gb) usually start with the first base of the LSC (regular sequence). However, some of them have irregular starting points, in which the first base of their non-LSC is the starting point.

The only previously released tool named IRscope (https://irscope.shinyapps.io/irapp/) (*Amiryousefi, Hyvonen & Poczai, 2018*) can realize the visualization of the tetrad junction sites of regular linearized genome sequences, but fails in the visualization of irregular sequences, or obtain wrong results that are difficult to be seen. Therefore, in this study, a universal visualization tool was then developed to facilitate the comparative analysis of the IR and single-copy region boundaries of different chloroplast genomes and provide a clear visualization of the IR contraction and expansion of chloroplast genomes.

## MATERIALS & METHODS

### Input data

The software input file can be complete chloroplast genome sequence files in .gb format or the configuration files of genome sequences. The .gb files should contain the annotation information, including gene tag and gene name (/gene = ""). Configuration files are two columns separated by Tab symbols. The first column is the files path in .gb format, and the second column is the corresponding junction site information, which can be used to customize the junction sites. The program read the .gb file and recorded the location information and name of each gene to save the sequence for subsequent IR region search.

### Search and adjustment of IR region

The starting point of a given linearized chloroplast genome sequence from GenBank may be located at any site on the circular genome. Hence, the exact junction site of the sequence where the starting point is located in the IR region is difficult to find. CPJSdraw adjusted the sequences as follow to more conveniently find the position of the IR region. First, the genome sequence was broken into subsequences called kmer with a size of 55bp and a step size of 1 bp; the frequencies of the occurrence of each kmer and its reverse complementary
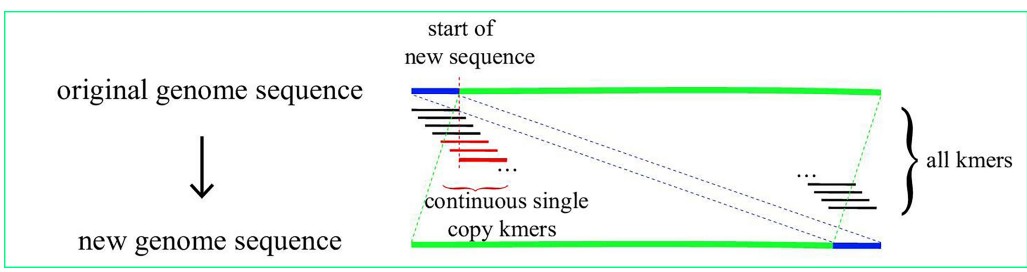

**Figure 1** Schematic view of IR region search and adjustment.

kmer were counted. A kmer that occurs only once is called a single-copy kmer. When a genome had $N$ consecutive single-copy kmers, the position of the $N$ kmer was considered the new starting point, and the sequence before this position was considered end point to form a new sequence. The starting point of the sequence was then relocated in the single-copy region, whereas the IR region was completely located on the new sequence and would not cross the starting and end points of the sequence (Fig. 1). CPJSdraw called mummer (*Kurtz et al., 2004*) to identify the IR region, and mapped the identified location to the original genome sequence to complete adjustment. Finally, CPJSdraw formatted the first base of the LSC as the starting point of the chloroplast genome.

## Selection and grouping of representative chloroplast genome

The chloroplast genome sequences tested were downloaded from the National Center for Biotechnology Information databases (https://www.ncbi.nlm.nih.gov/genome/browse/). According to the analysis of all downloaded chloroplast genomes, if a genome contained IR and IR contained at least one gene, combined with that the tRNAs are short in length, and were generally < 100 bp bp in these genomes, the filter threshold was then set to < 100 bp. Thus, by identifying the longest IR sequences of chloroplast genomes, if the longest IR of the genomes were < 100 bp, those genomes were filtered, and the genomes without annotation were also filtered. The obtained genomes were divided into the following nine groups: a. genomes with a starting point in the first base of the LSC; b. genomes with a starting point in the LSC; c. genomes with a starting point in the first base of the IRb; d. genomes with a starting point in the IRb; e. genomes with a starting point in the first base of the SSC; f. genomes with a starting point in the SSC; g. genomes with a starting point in the first base of the IRa; h. genomes with a starting point in the IRa; and i. genomes with an IRa, IRb, or SSC region, and having only one gene. Two representative chloroplast genomes were randomly selected from each group, and a total of 18 representative chloroplast genomes were selected for analysis. All the representative chloroplast genomes tested are also available at GitHub: https://github.com/xul962464/CPJSdraw for English users.

## Visualization and comparison of the junction sites

CPJSdraw called the SVG module in Perl, first plotted basic information, such as species name, sequence length, four regions, and region size; analyzed the junction sites of the four regions of the linearized chloroplast genome sequences; and displayed the visualization
result using /organism ="" in the .gb file. The sequences were displayed in "LSC-IRb-SSC-IRa-LSC" format, showing only two genes located on both sides of the junction site and indicating the lengths and distances of the genes from the junction sties. The number of genes in each area was evaluated. No plotting was done if no genes. If a gene was present, it was plotting in the middle of the region, and the starting and ending points of the region were indicated. If two or more genes were present in the region, the first and last genes were plotted according to their location. The transcription directions of genes were represented by arrows to avoid possible overlap of the genes during plotting. Right arrow indicated that the genes were in the positive chain, and left arrow indicated that the genes were in the negative chain. The visualization images from CPJSdraw are provided in three formats, namely, .svg, .pdf, and .png, for downloading.

The selected chloroplast genomes were tested simultaneously using the previously released software IRscope (*Amiryousefi, Hyvonen & Poczai, 2018*). The resultant visualization images were provided for comparison with those obtained from CPJSdraw.

## RESULTS

### Statistics of tetrad junction sites
A total of 8,487 (as of 2022/6/1) complete chloroplast genomes were downloaded from NCBI, and 8,229 were obtained after filtering in this study. Among them, 8,194 were circular genomes, and 35 were linear genomes. The first base of the LSC was the most common starting point (72.33%) in the 8,229 linearized genomes, followed by starting points located in the LSC and IRb (13.04% and 13.01%, respectively) (Table 1).

### Analysis and comparison of junction sites
Nine representative groups of genomes were obtained and analyzed according to the different starting points of the linearized genome sequences and the different number of genes contained in a certain region, and each group contained two genomes. All of the tetrad region locations of the representative chloroplast genomes are listed in Table 2.

### Visualization and comparison of the junction sites
Results showed that CPJSdraw could analyze and visualize the junction sties of all the chloroplast genomes tested (Fig. 2). For all the cases, CPJSdraw gave a very clear display of junction sites and indicated the contraction and expansion of the IR regions of the tested chloroplast genomes. In comparison, the plot generated by IRscope mainly has the following problems: (1) although IRscope could visualize groups a and b, information, such as all the genes at the junction sites (for example, the gene on the left closed to the junction of SSC/IRa in group a of Fig. 3), the direction of gene transcription (all genes in Fig. 3), and the distance (for example, *rpl22* and *ndhF* in group a of Fig. 3) from the junction sites were not fully displayed (Fig. 3); (2) although group h could be visualized, the LSC region became longer and the IR region became shorter, because a small part of the IRb sequence was mistaken for the LSC sequence (Fig. 3); (3) some gene names were overlapped (for example, group b in Fig. 3); (4) the most regrettable is that,, except for groups a, b, and h, the visual results of the other six groups could not be obtained by online

**Table 1 Analysis of starting point locations of 8,487 linearized chloroplast genomes.**

| Starting locations | Numbers | Percentage (%) |
|---|---|---|
| Start with LSC | 5,952 | 72.33 |
| Start in LSC | 1,073 | 13.04 |
| Start with IRb | 15 | 0.18 |
| Start in IRb | 36 | 0.44 |
| Start with SSC | 19 | 0.23 |
| Start in SSC | 58 | 0.70 |
| Start with IRb | 5 | 0.06 |
| Start in IRb | 1,071 | 13.01 |
| Total | 8,229 | 100.00 |

IRscope, and the local version of IRscope also failed. The reason for the failure could be due to the abnormal starting point or short IR region of the linearized genomes, leading to the failure to identify the junction site.

## DISCUSSION

Plant chloroplast genomes usually have a collinear sequence. The IR regions of genomes have important roles in conserving essential genes and stabilizing structure because of their relative conserved structure, compact size, and maternal inheritance. Thus, IR expansion and contraction are considered useful for the assessment of plant genetic diversity (*Jansen et al., 2005*; *He et al., 2020*). For example, IR expansion in the chloroplast genomes of *Pothos scandens* and its relatives decreased the evolutionary rate of protein-coding genes that shifted from the SSC region to the IR region, whereas an increase in evolutionary rate was observed in the genes that transferred from the IR region to the LSC region (*Abdullah Henriquez et al., 2020*).

IR contraction and expansion may lead to gene loss or gain, which are the main causes of the size variations in chloroplast genomes. The lost or gained genes are usually *rps19*, *rpl22*, *ycf1*, *ycf2*, *ndhF*, and *trnH*, which often resulted in their corresponding pseudogenes, especially for *rps19*, *ndhF*, and *ycf1* (*Zhu et al., 2016*). For instance, the *rpl22*, *ndhF*, *ycf1*, *rps19*, and *trnH* genes were detected at the junctions among IR, LSC, and SSC regions of the chloroplast genomes of *Salix* species, in which, *rps19* and *ycf1* were the lost or gained genes (*He et al., 2022*). Analysis of the IR regions of 14 *Paphiopedilum* species showed that the transferred genes were *rps19*, *ycf1*, *ndhD*, *psaC*, *rps15*, *ccsA*, and *trnL* (*Guo et al., 2021*). Besides, analysis of the IR regions of the chloroplast genomes of 34 monocotyledon Triticeae species showed that the lost or gained genes were *rps19*, *rpl22*, *rps15*, and *ndhH* (*Chen et al., 2021a*; *Chen et al., 2021b*). Analysis of the IR regions of the chloroplast genomes 30 representative chloroplast genomes in Arecaceae showed that the lost or gain genes were *rps19*, *ycf1*, *ndhF*, and *rpl32* (*Chen et al., 2022*). Even for seven fern species, analysis of the IR regions of chloroplast genomes showed that the lost or gain genes were *ndhF*, *trnI*, *chlL*, and *ndhB* (*Fan et al., 2021*).

In the present study, the analysis of 8,229 linearized chloroplast genomes downloaded from NCBI, revealed that a small part of chloroplast genomes starts in the IR region instead

**Table 2** Tetrad structures of chloroplast genomes.

| Group | Accession No. | Species name | Region location |
|---|---|---|---|
| a | NC_036102.1 | *Sophora alopecuroides* | LSC:1–84221, IRb:84222–110095, SSC:110096–128234, IRa:128235–154108 |
| | NC_056151.1 | *Sophora moorcroftiana* | LSC:1–83342, IRb:83343–107133, SSC:107134–125139, IRa:125140–148930 |
| b | NC_062457.1 | *Zingiber teres* | LSC:163399–88112, IRb:88113–117864, SSC:117865–133646, IRa:133647–163398 |
| | NC_062475.1 | *Solanum velardei* | LSC:155437–86008, IRb:86009–111535, SSC:111536–129909, IRa:129910–155436 |
| c | NC_038203.1 | *Machilus pauhoi* | LSC:58951–152621, IRb:1–20074, SSC:20075–38876, IRa:38877–58950 |
| | NC_038204.1 | *Machilus thunbergii* | LSC:58901–152551, IRb:1–20050, SSC:20051–38850, IRa:38851–58900 |
| d | NC_053720.1 | *Colobanthus nivicola* | LSC:58978–142328, IRb:142329–16445, SSC:16446–33651, IRa:33652–58977 |
| | NC_053721.1 | *Colobanthus lycopodioides* | LSC:58937–142581, IRb:142582–16439, SSC:16440–33616, IRa:33617–58936 |
| e | NC_057956.1 | *Camellia achrysantha* | LSC:44327–130575, IRb:130576–156658, SSC:1–18243, IRa:18244–44326 |
| | NC_057957.1 | *Camellia chrysanthoides* | LSC:44332–130895, IRb:130896–156959, SSC:1–18267, IRa:18268–44331 |
| f | NC_048463.1 | *Wolffia globosa* | LSC:45381–137551, IRb:137552–169361, SSC:169362–13570, IRa:13571–45380 |
| | NC_050170.1 | *Cyperus rotundus* | LSC:47738–148698, IRb:148699–186119, SSC:1–10315, IRa:10316–47737 |
| g | NC_050999.1 | *Heterotis rotundifolia* | LSC:26740–112476, IRb:112477–139215, SSC:139216–156336, IRa:1–26739 |
| | NC_056142.1 | *Verbena officinalis* | LSC:25809–110326, IRb:110327–136134, SSC:136135–153491, IRa:1–25808 |
| h | NC_062509.1 | *Solanum caripense* | LSC:2–85726, IRb:85727–109824, SSC:109825–130908, IRa:130909–1 |
| | NC_060516.1 | *Cymbidium cyperifolium* | LSC:688–85961, IRb:85962–111550, SSC:111551–127763, IRa:127764–687 |
| i | NC_053746.1 | *Rhododendron platypodum* | LSC:1–109134, IRb:109135–153784, SSC:153785–156397, IRa:156398–201047 |
| | NC_061648.1 | *Larix kongboensis* | LSC:91317–34931, IRb:34932–35367, SSC:35368–90880, IRa:90881–91316 |

**Notes.**

Note: The positions for the regions are represented from the starting point to the ending point. A starting point greater than the ending point, indicates that the region spans the beginning and end of the sequence.

of the first base of the LSC. Regardless of where linearized chloroplast genomes start, CPJSdraw could found the junction sites of the four regions of chloroplast genomes, and the results were detailed and accurate. In comparison, the previously released software (*Amiryousefi, Hyvonen & Poczai, 2018*) may lead to wrong results or even no results because it does not consider the regular starting point of chloroplast genomes.

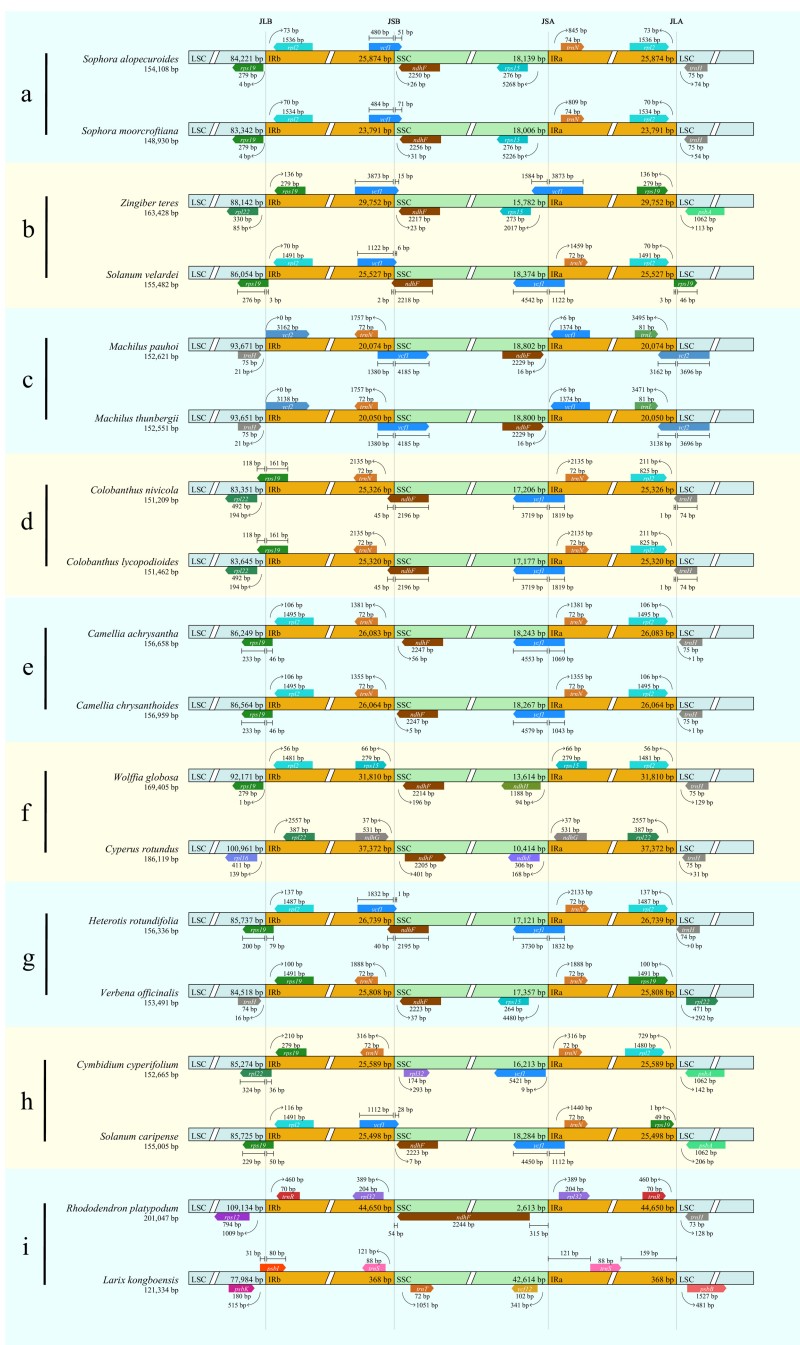

**Figure 2** **Visualization of the junction sites of the representative chloroplast genomes based on CPJS-draw analysis.** Note: (A–I) represent the nine groups of chloroplast genomes.

## CONCLUSIONS

A previously released tool IRscope for the visualization of junction sites of chloroplast genomes does not consider diversity of the starting point of genomes, which leads to
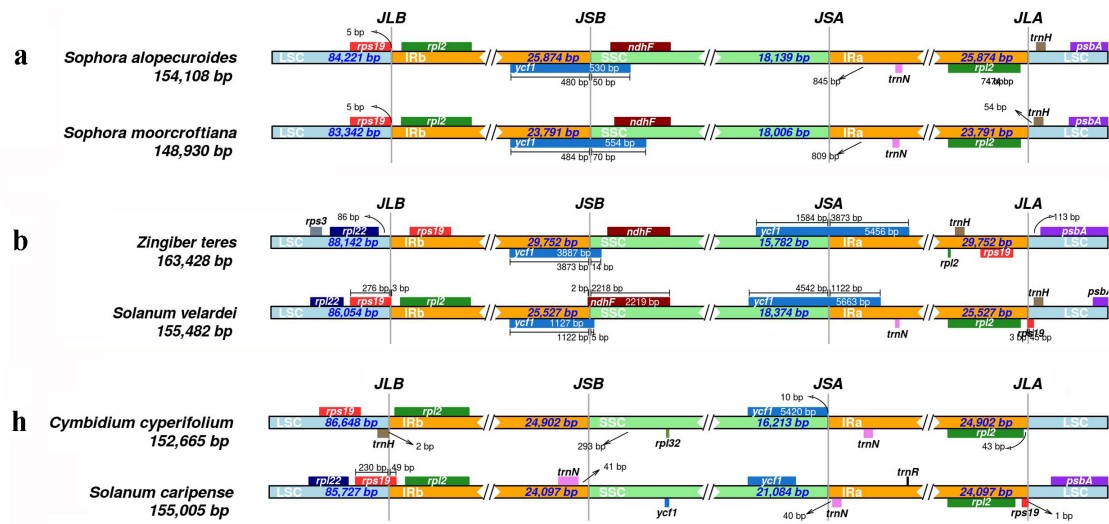

**Figure 3** Visualization of the junction sites of the representative chloroplast genomes based on IRscope analysis.

incorrect results or even no results. In this study, a new tool named CPJSdraw was developed for visualizing the junction sites of genomes. CPJSdraw can format the starting point of irregular linearized genome, correct the junction sites of IR and single-copy regions, display the tetrad structure, visualize the junction sites of any number (≥1) of chloroplast genomes, show the transcription direction of genes adjacent to junction sites, and indicate the IR expansion or contraction of chloroplast genomes. Therefore, CPJSdraw is a software that is universal and reliable in analysis and visualization of IR expansion or contraction of chloroplast genomes. CPJSdraw has more accurate analysis and more complete functions when compared with IRscope.

# ACKNOWLEDGEMENTS

The authors would like to thank Ali Amiryousefi, Jaakko Hyvonen, Peter Poczai for developing the previously released tool IRscope.

## Funding

This research was supported by the National Natural Science Foundation of China (31960320 and 32260415). The funders had no role in study design, data collection and analysis, decision to publish, or preparation of the manuscript.

## Grant Disclosures

The following grant information was disclosed by the authors:
The National Natural Science Foundation of China: 31960320, 32260415.

## Competing Interests

Lei Xu, Haidong Gao, Lei Liu, and Xiangyang Zhou are employed by Nanjing Genepioneer Biotechnologies Co., Ltd. The authors declare there are no competing interests.

## Author Contributions

- Huie Li conceived and designed the experiments, performed the experiments, prepared figures and/or tables, authored or reviewed drafts of the article, and approved the final draft.
- Qiqiang Guo performed the experiments, prepared figures and/or tables, authored or reviewed drafts of the article, and approved the final draft.
- Lei Xu conceived and designed the experiments, performed the experiments, prepared figures and/or tables, authored or reviewed drafts of the article, and approved the final draft.
- Haidong Gao analyzed the data, prepared figures and/or tables, authored or reviewed drafts of the article, and approved the final draft.
- Lei Liu analyzed the data, prepared figures and/or tables, authored or reviewed drafts of the article, and approved the final draft.
- Xiangyang Zhou analyzed the data, prepared figures and/or tables, authored or reviewed drafts of the article, and approved the final draft.

## Data Availability

The code of CPJSdraw as a perl package and tested data are available at GitHub and Zenodo: https://github.com/xul962464/CPJSdraw.

xul962464. (2023). xul962464/CPJSdraw: v1.0.0 (v1.0.0). Zenodo. https://doi.org/10.5281/zenodo.7669480

An online version of the software is available with a Chinese interface at Gene Pioneer. This requires a user to log in: http://cloud.genepioneer.com:9929/#/tool/alltool/detail/335.

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
