# Peer review of "CPJSdraw: analysis and visualization of junction sites of chloroplast genomes"

_PeerJ, doi:10.7717/peerj.15326_

## Round 0.1 · original submission · Minor Revisions

Dear Dr. Li

The reviewers have requested minor revisions of the manuscript. Kindly revise the manuscript as per the comments.

Overall the manuscript is well written and described all the analysis very clearly.
Thank you.

·

Basic reporting

The link to the Github repository should be placed into the abstract also. Most readers read the abstract only.
L41: I suggest to highlight to role of IR region, why it’s important. If its function limited to diversity only?
Speaking of style, I don't feel qualified to judge about the English language and style, but most sentences are easy to understand.

Experimental design

L65: Does the CPJSDraw work with contigs?
L88, L94: What the criteria of 18 genomes selection?
L120: How the filtering was performed?
L68: the “merge.cfg” at repository contains incorrect folder names and absolute paths. Please prepare the sample dataset to ready to use state.
Is it possible to change the default font? I’ve received a “WARNING: Font 'Times New Roman,italic,400' not found. Substituting with default font.”

Validity of the findings

The developed software is ready to use and verified on number of genomes. Unfortunately, some errors in the Github repository should be corrected. The main Perl file runs even with invalid config file parameter (without error).
My notes about the main text:
• L37: if IR region length is “ranges from 20 kb to 27 kb”, how it * with “where the IR region exceeds 75 kb” (L40)?
• L170, L180: Sentences are duplicated.

Reviewer 2 ·

Basic reporting

The manuscript by Li et.al. presented the new tool CPJSdraw for visualizing IR junctions in chloroplast genomes. It showed improvement comparing to previously released IRscope through correction of starting points in irregular linearized sequences, and better display of gene direction and distance from junctions.

Pease find below a few minor comments. Hope the authors could address before considered for publication:
1. Could the authors provide more details on the search and adjustment of IR region? I.e. what was the range of K used in the kmer scanning process? Would the K length affect downstream result or computing speed?
2. The genomes with IR regions < 100bp were filtered before the analysis. Could the authors provide the rationale behind the threshold selection? Would CPJSdraw require specific chloroplast genome structure to function properly (i.e. require all IRa/IRb/LSC/SSC to be present, and/or require minimum lengths for each component)?
3. Please consider flip the order of the two genomes in each group in Fig.3 to match that in Fig.2 for easier comparison. And if possible, please highlight the differences in the plot to clearly show the improvement in CPJSdraw against IRscore.

Experimental design

no comment

Validity of the findings

no comment

---

## Round 0.2 · accepted · Accept

The authors have addressed all of the reviewers' comments.
I am writing to inform you that your manuscript - CPJSdraw: Analysis and visualization of junction sites of chloroplast genomes - has been Accepted for publication. Congratulations!!

·

Basic reporting

pass

Experimental design

I still recommend to prepare ready-to-use Gitgub/Gitlab repository, not Figshare/Zenodo file. The software is constantly developing (I see 25 commits already in https://github.com/xul962464/CPJSdraw) and after years users will see the current version.
Please test your software by inexperienced users, they often find errors.

Validity of the findings

Everything fine, pass.

Reviewer 2 ·

Basic reporting

The authors have addressed the comments very well, and revised the draft accordingly. I am satisfied with the revision and have no further comments.

Experimental design

no comments

Validity of the findings

no comments